# Development of Crystallinity of Triclinic Polymorph of Tricalcium Silicate

**DOI:** 10.3390/ma13173734

**Published:** 2020-08-24

**Authors:** Simona Ravaszová, Karel Dvořák

**Affiliations:** Faculty of Civil Engineering, Brno University of Technology, Veveří 331/95, 602 00 Brno, Czech Republic; ravaszova.s@fce.vutbr.cz

**Keywords:** triclinic tricalcium silicate, alite, crystallite size, Scherrer equation, growth rate

## Abstract

Tricalcium silicate phase is one of the main components of modern Portland cements. One of the major industrial challenges in the field of cement production is mapping the influence of individual clinker minerals and their polymorphs on the properties of industrially produced clinkers. The primary goal of this work is to improve the fundamental knowledge of understanding the process of alite formation and development from a crystallographic point of view. This study focuses on the observation of the crystallization process of triclinic alite during the firing process, which to date has not been thoroughly described. The effects of a wide range of temperatures and sintering periods on crystallinity were assessed on samples fired in platinum crucibles in a laboratory furnace. X-ray analysis—together with calculation of crystallinity using Scherrer’s equation—was used for observing the crystallite size changes of T1 alite polymorph. According to the acquired results, among the most technologically and economically advantageous regimes of production of a high-quality triclinic alite is the temperature of 1450 °C and sintering time of two hours. The most significant changes in the crystallite size occurred within the first hour of sintering for the whole investigated temperature range.

## 1. Introduction

The modern trends in industry and commerce are towards environmental sustainability and economic savings. This approach has led to the development of numerous enhancements throughout various industrial branches—from medicine and bioengineering [1], metallurgy and materials engineering [2], recycling and recovery [3], thermal engineering [4], to civil engineering and cement industry [5]. Research studies within the latter branch have paid nonnegligible attention to sustainable development, since—in response to long-time market demands—alite contents in Portland cements have been gradually increasing. However, the manufacture of cements with higher alite contents brings about higher environmental impacts. Methods for reducing emissions are being continuously developed in the cement industry, since the production process of Portland cement produces significant amount of CO_2_, which is released during both, the thermal decomposition of limestone carbonates and fuel combustion. By this reason, refined fuels are being gradually substituted by alternative fuels, which introduces economic and environmental benefits, but brings about many technological issues, too. These issues originate from increasing the contents of certain elements; the effects of the changes on the properties of the produced clinkers are subject to continuous investigations and monitoring [6,7,8,9,10,11]. Another possible way how to reduce CO_2_ emissions is to adjust the raw-meal composition or decrease the clinker firing temperature. Manufacturing of cements with reduced environmental impacts would result in decreasing the overall environmental footprint of concrete as a construction material.

The main contemporary issue of industrial production of cements is the variability of their mineralogical composition, which depends on the presence of impurities (originating from raw materials or fuel) and the firing process. Therefore, the process of synthesis of clinker minerals is crucial for determination of their subsequent behavior in industrially produced cements. In order to characterize and possibly influence the variability of mineralogical composition of clinkers, the effects of individual polymorphs of clinker minerals on the properties of industrially produced clinkers, especially from the viewpoints of acting chemistry and their internal structures, need to be studied [12,13,14]. Generally, about twenty individual minerals can be identified within clinkers, but four of them are the most significant. These are tricalcium silicate (C_3_S, alite), dicalcium silicate (C_2_S, belite), tricalcium aluminate (C_3_A) and tetra calcium aluminate ferrite (C_4_AF). These four clinker minerals are the primary ones determining the properties of the cements (together, they form over 90% of the total mass of clinkers). Given by their characteristic properties, each of these minerals affects the properties of the individual cement specifically. Except for tetra calcium aluminate ferrite, all the major clinker minerals exhibit strong polymorphism [15].

Most of the standard methods used to prepare clinker minerals are rather time-consuming; the solid-state reaction and sol–gel method are the most widely used procedures. The main limiting and time-consuming processing steps/factors during preparation of clinker minerals are disc pressing, demanding sintering time and limited capacity of the used platinum crucibles and furnaces. The composition of the raw material and the setting of the firing parameters are the primary affecting factors determining the final phase composition of the clinker [16,17]. However, another important factor influencing the phase composition of the clinker, quite a negligible attention to which has been paid so far, is mechanical–chemical activation. The possible presence of active surfaces can influence the process of C_2_S formation during the solid-state reaction and consequently facilitate the process of C_3_S formation from the melt. By using suitable activation grinding technology and by optimizing the milling process, it should be possible to avoid disc pressing, reduce the firing temperature and sintering time and promote the formation of the desired polymorphs [18,19].

The most important clinker mineral is tricalcium silicate (C_3_S), which is characterized by extensive polymorphism; seven structural modifications of C_3_S have been identified so far: three triclinic (T1, T2, T3), three monoclinic (M1, M2, M3) and one trigonal (R) [15,17]. The sequence of alite transformations is shown in the following Figure 1 [15]:

The first mention of polymorphism of the C_3_S dates back to the early 1950s. Jeffery documented that pure C_3_S consists of triclinic (T1) and monoclinic (M3) modifications of alite and further developed a hypothesis for the high-temperature polymorphic form with trigonal structure [20]. The first systematic work on the topic was done by Regourd, Guiniere and their co-workers [21,22,23]. Regourd pointed out that each polymorphic alite form could be identified at a certain angular range. Subsequent study was performed by Golovastikov and his team, who investigated the T1 structure. Later on, within a few years, other structures were identified (given by the progress in the X-ray diffraction analysis methods) [24]. The advancements in equipment and software led to the introduction of other alite structures by Nishi et al. [25], Mumme [26] and Noirfontaine [27]. The most common polymorphs found in industrially produced clinkers are M1 and M3; M1 and M3 monoclinic polymorphs were studied by Maki et al. [28,29], who pointed out the possible influence of sulfate and magnesium impurities on stabilization of the particular polymorphs. This topic was also examined by Staněk and Sulovský [16].

The process of formation and stability of the individual alite polymorphs are the main factors influencing the technological properties of clinkers. By this reason, study of these factors is an important issue. This study presents fundamental research of the basic processes occurring during the processes of formation and development of triclinic alite from a microscopic point of view. The presented study focuses on monitoring of the development of crystallite size during the firing process of a T1 polymorph of alite; T1–alite was chosen because it can be relatively easily prepared by cooling pure Ca_3_SiO_5_ compound.

## 2. Materials and Methods

The process of mechanical activation with high-temperature solid-state reaction was used to synthesize the triclinic alite. The basic compounds of calcium carbonate (CaCO_3_, p.a. purity, Lach-Ner, Neratovice, Czech Republic) and silica (SiO_2_, p.a. purity, Lach-Ner, Neratovice, Czech Republic) were dosed in the amounts corresponding to the molar ratio of tricalcium silicate components (73.6% CaO and 26.3% SiO_2_), see the Table 1.

To prepare the raw meal, the semi-wet approach was applied [30]. Calcium carbonate with silica oxide in the molar ratio of 3:1 were weighed and mechanically activated in a planetary mill (Pulverisette 6, Fritsch, Idar-Oberstein, Germany). The water/powder ratio of 0.8 was used. The volume of the hardened steel milling capsule, 25 grinding steel balls with 20 mm in diameter in which were inserted, was 0.5 dm^3^. The grinding time was 20 min, and the grinding speed was 500 rpm. The dosage of the raw material mixture was 1000 g in total. The mechanically activated raw material mixture was dried in a laboratory dryer (Binder C170, Binder GmbH., Tuttlingen, Germany) at 105 °C for 24 h. During drying, nodules of 10 mm in diameter formed naturally from the powder mixture. The nodules were dosed directly into 30-mL platinum crucibles prepared for the subsequent firing process.

For the solid-state reaction procedure, a ventricular high-temperature kiln (2017S, Clasic CZ s.r.o., Řevnice, Czech Republic) with Kanthal super heating elements was used. The solid-state reaction was carried out at six different temperatures of 1350, 1400, 1450, 1500, 1550 and 1600 °C and six different sintering periods of 0, 1, 2, 3, 4 and 5 h. The heating rate was 10 °C/min. At the end of each firing mode, the crucibles were removed from the furnace and immediately cooled by a stream of cold air. The product of this production step in the form a block was subsequently milled to a powder in a vibratory disc mill (RS 200, Retsch, Haan, Germany) at 900 rpm for 20 s.

The XRD analysis was performed with a multifunctional diffractometer (XRD, Empyrean, PANalytical B.V., Almelo, The Netherlands) with Cu anode; Kα was used as the radiation source with the parameters of: λ = 1.540598 Å, accelerating voltage 45 kV, beam current 40 mA, diffraction angle 2θ in the range from 5° to 90° with a scan-step of 0.01°. The ICSD database (released 2012) was used to qualitatively analyze the diffraction patterns. HighScore plus software (3.0e, PANalytical B.V., Almelo, The Netherlands) was used to identify the individual phases, perform their quantification and determine the amount of the amorphous phase, which was estimated using the “constant background intensity” method, the crystallinity of a sample in which is defined as the intensity ratio of the diffraction peaks and of the sum of all the measured intensities (can be calculated using Equation (1)):(1)$$C=\frac{100\cdot \sum I}{\sum I_{tot}-\sum I_{const.bgr.}}$$
where *C* is crystallinity in %, *I* is area of crystalline peaks, *I_tot_* is total area and *I_const.bgr._* is area of constant background. The constant background intensity is subtracted from the total intensity [31].

The size of the crystallites was evaluated on a selected diffraction line at an appropriately selected crystallographic plane positioned in a major crystallographic direction. We selected a high-intensity diffraction line which covered the selected plane and, at the same time, did not overlap with other diffraction lines (especially beta C_2_S), defined by the “hkl” Miller index of $4\bar{4}5$. Calculation of the size of the crystallite was based on the measurement of FWHM (full width at half maximum) and the Scherrer equation [32] with Warren correction [33] (Equation (2)),
(2)$$L=\frac{K\cdot \lambda }{\cos\theta }.\frac{1}{\beta }=\frac{K\cdot \lambda }{\cos\theta }.\frac{1}{\sqrt{B^{2}-b^{2}}}$$
where *L* = crystallite size, *K* = Scherrer constant (the value of 0.89 was used), *λ* = K_α1_ (wavelength of X-ray radiation), *θ* = diffraction angle, *B* = FWHM (full width at half maximum), *b* = FWHM of the size/strain standard used (lanthanum hexaboride).

The instrument extension was set to the LaB_6_ standard (lanthanum hexaboride). LaB_6_ is considered to be a fully crystalline material, the size of the crystallites in which is theoretically equal to infinity. Peaks of LaB_6_ were determined in the positions identical to the selected diffraction lines in the same way in the HighScore plus software. The FWHM values of LaB_6_ were calculated from the equation depicted in Figure 2. Other factors affecting the accuracy of this evaluation method, such as strain, were neglected since they were not crucial for the aim of this experiment—to assess the trend of development of the size of the crystallites during the firing process.

## 3. Results

Qualitative monitoring of the mineralogical composition was performed by X-ray analysis. Beside T1–tricalcium silicate, residual dicalcium silicate (beta C_2_S) (ICSD 98-007-9552), free lime (f-CaO) (ICSD 98-006-0704) and portlandite (Ca(OH)_2_) (ICSD 98-005-3989) were identified on the X-ray scans. The amount of Ca(OH)_2_ was converted to free CaO by the loss on ignition method; subsequent recalculation of the contents of individual minerals followed. The dependence of the contents of minerals on the temperature and sintering period is shown in Table 2 and graphically in Figure 3.

The observed and calculated amount of the amorphous phase was less than 1–2% in all samples. Based on the results of the X-ray analysis, the following statements can be done. With increasing firing temperature, the amount of triclinic alite increases linearly from 9.1% at 1350 °C, to 99.4% at 1600 °C. Identical phenomenon was observed for the increased sintering periods at the individual temperatures. The most significant time-dependent difference in the content of alite was observed at the lowest temperature of 1350 °C; after 5 h of sintering, the content of alite increased from 9% to 90%. The higher was the firing temperature, the smaller was the observed difference in the content of alite between the individual sintering periods.

At the temperature of 1350 °C, the content of triclinic alite was approximately 89% after 5 h of sintering. Similar amounts of alite were achieved after the sintering periods of 2 or 1 h for the higher temperatures of 1400 °C and 1450 °C. Since the aim of the study was to acquire the highest possible amount of alite at the lowest possible temperature and sintering time period, the temperature of 1350 °C appears to be insufficient. The highest alite contents of 99.1%, 99.6% and 99.4% were acquired at the following three temperatures (and corresponding sintering time periods), respectively: 1450 °C (2 h), 1500 °C (4 h) and 1600 °C (5 h). The belite contents were equal to 0%, and the amounts of free lime were negligible (<1%). Therefore, the temperature of 1450 °C and 2 h of sintering seems to be the ideal firing regime from the viewpoint of reasonable sintering time and economic complexity.

Due to the complexity of the crystal lattice of triclinic alite, a crystallographic plane characterized with a dominant crystallographic direction, and, at the same time, manifesting in the diffraction record with a significant isolated peak (without overlapping with the belite line) was chosen. The crystallographic plane $4\bar{4}5$ is manifested in the diffraction pattern by a peak at the 2θ angle of 34.291°, see Figure 4.

Scherrer’s equation was used to monitor the dependence of crystallite size on the temperature and sintering time period on the $4\bar{4}5$ crystallographic plane; see Table 3 for the values of sizes of the crystallites and Figure 5a–d for graphic presentation. Based on the development of crystallinity during the firing regimes, the growth rate of crystallites was evaluated when the individual temperatures were reached, see Table 4.

As can be seen in Figure 5, with increasing temperature, the intensity of the diffraction lines increased, and the shapes of the diffraction lines became narrower and longer. Similar increase in intensity was observed for a single sintering temperature (specifically 1450 °C), however, with increasing sintering time period (Figure 5b). The most significant changes in crystallinity occurred within 1, possibly 2, hours of sintering, when tricalcium alite is still forming. This phenomenon is the most evident in the detailed section of the graph depicted in Figure 5d; as can be seen, steady increase in the crystallite size occurred with increasing temperature up to 1 h of sintering (see Figure 5c,d). After 1 h of sintering, the crystallite size remained more or less constant.

As regards the growth rate of the crystallites, the highest rate of crystallite formation was observed within just 1 h. In addition, the higher was the temperature, the faster was the crystallite growth rate. After 1 or 2 h of sintering, the growth rate decreased, and the crystallite size was more or less constant.

## 4. Discussion

The present study deals with monitoring of the crystallization process of triclinic alite during the firing process. The parameters of preparation of the raw material were optimized to maximize the amount of alite in the sample while maintaining reasonable processing time. Triclinic alite was produced by mechanical activation of a mixture of the raw material in a high-energy mill in a water environment. During drying of the material ground in the water environment, the slurry had a tendency to agglomerate to solid nodules (this method of preparation was designed according to the semi-dry method of preparation of raw meal used in the clinker industry). This simplifies the technological process of preparation without disc pressing, which is the most common way of preparing the material for the solid-state reaction procedure. Selecting a suitable composition of the mixture of the raw material subjected to high-energy milling enabled preparation of a high-quality triclinic alite in the entire range of selected temperatures and sintering periods, as demonstrated in Table 2.

With increasing firing temperature and sintering period, the amount of alite increased—relative to other minerals—as demonstrated in Figure 3a,c. Traces of portlandite were identified in all the samples; the presence of this mineral in the samples originated from rapid hydration of unreacted free lime while waiting for being analyzed by the X-ray diffractometer. Samples fired for a short time periods (0–30 min) contained high amounts of free lime, which tended to quickly hydrate with air humidity to form portlandite. Since portlandite is a product of CaO only, this is a very fast reaction. We further performed recalculation of the real amount of free lime in the samples by the loss of water molecule of the portlandite and increased the amounts of the remaining minerals by this value. The results showed that all the raw materials reacted to the crystalline phase when the temperature of 1350 °C was reached. The temperature of 1350 °C is sufficient for all the components to occur in the crystalline form (alite, belite, lime, etc.), because the amount of the amorphous phase was lower than 1–2% in all cases. Therefore, we can state that the contents of individual minerals in the samples are not affected by the amorphous content.

High-quality T1 alite (99.1% C_3_S, 0.9% CaO) was acquired at 1450 °C and two hours of sintering, see Table 2. The Scherrer method was used to assess the effects of the firing process on the development of the size of the crystallites of triclinic alite. This method was chosen due to the complexity of the triclinic structure, difficulty of fit and significant peak overlapping of the individual crystallographic planes—especially with beta belite. Generally, the most significant changes in the crystallite size occurred within one, possibly two hours of sintering for the whole temperature range, as shown in Figure 5a,d. With increasing firing temperature, the size of the crystallites increased continuously. While at the temperature of 1350 °C the size of the crystallites was 33 nm, at the temperature of 1550 °C the size of the crystallites was 51 nm, see Table 3. With the increase in the sintering period to one hour, the size of the crystallites increased at all the temperatures. The higher the temperature, the smaller the difference in the crystallite size.

We consider that the time of one hour of sintering is critical. The average value of the size of the crystallites at this time was around 52 nm, which is 95% of the maximum size. After one hours, we no longer observed significant changes; the size of the crystallites was constantly around 50 nm, see Table 3. The growth rate of crystallites had a linear trend, as depicted in Figure 6. As the firing temperature increased, the crystallites grew faster. At 1350 °C, the growth rate was 0.88 nm/min, at 1600 °C it was as high as 1.37 nm/min, as seen in Table 4. This fact could be explained by the different concentrations of the crystallization nuclei, which are higher at higher temperatures. Prolonging the sintering time period at the individual temperature resulted in a slight decrease in the crystallites growth rate, which can be attributed to the fact that the size of the crystallites did not significantly change after one (or two) hours of sintering.

Triclinic alite was prepared within the entire investigated temperature range and at all the sintering time periods. Within the technologically and economically most advantageous variants of production of a high-quality triclinic alite is the temperature of 1450 °C with two hours of sintering, the size of the crystallites at which reaches the value of 52 nm and the growth rate is around the value of 1.03 nm/min.

## 5. Conclusions

The main aim of this work was to describe the development of the crystallization process of triclinic tricalcium silicate. Based on the acquired results, the following can be stated:
▪Triclinic alite manufactured at 1350 °C is of insufficient quality;▪The most significant changes in the crystallite size occurred during 1, possibly 2, hours of sintering within the whole investigated temperature range;▪The average value of the crystallite size at this sintering time is around 52 nm, which is 95% of the maximum size;▪After one hours, the size of the crystallites settled around 50 nm, and no significant changes were further observed;▪The growth rate of crystallites has a linear trend; as the firing temperature increases, the crystallites grow faster;▪The firing regime of 1450 °C, two hours of sintering and 1.03 nm/min growth rate seems to be ideal from the viewpoint of time and economic complexity.

The fact that the fundamental nature of this research belongs to the Basic Research needs to be emphasized. The goal of the project is to provide new knowledge, information and connections related to understanding the formation and development of individual clinker minerals from a microscopic point of view. The study of the origin, stability, and properties of various calcium silicate polymorphs is important to better understand the processes involved in the production of Portland clinker. Newly available technologies can contribute to reduction in the energy consumption of firing and reduce CO_2_ emissions. At the same time, the synthetic pure clinker phases can be subject to further investigations (hydration process, effects of plasticizers, etc.).

## Figures and Tables

**Figure 1 materials-13-03734-f001:**
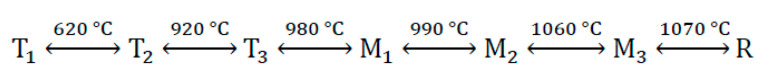
The sequence of alite transformations.

**Figure 2 materials-13-03734-f002:**
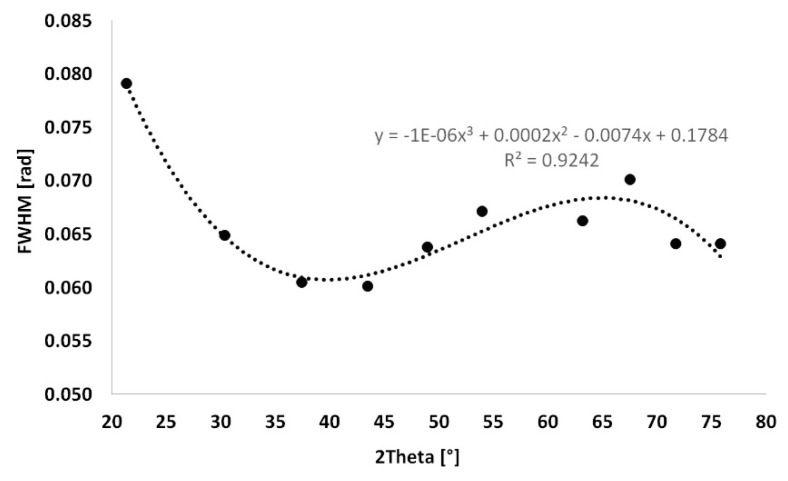
Instrument extension of LaB_6_.

**Figure 3 materials-13-03734-f003:**
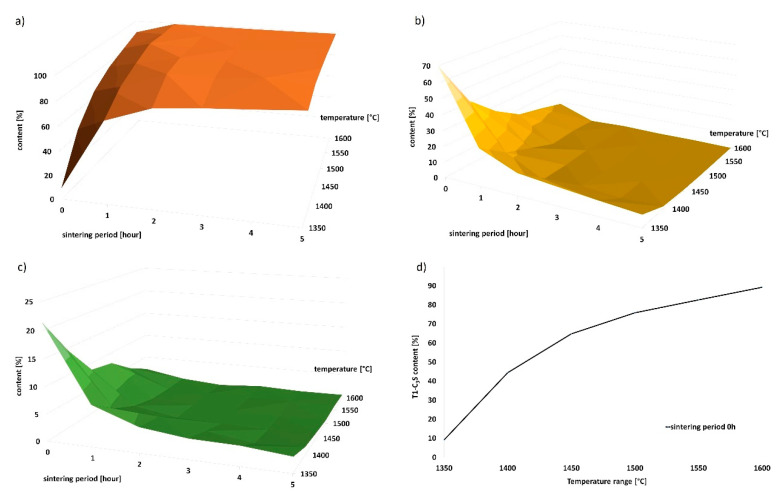
Graphic representation of change in mineral composition with increasing sintering period. (**a**) T1–C_3_S content; (**b**) β-C_2_S content; (**c**) f-CaO content; (**d**) linear dependence of the amount of T1–C_3_S on increasing temperature.

**Figure 4 materials-13-03734-f004:**
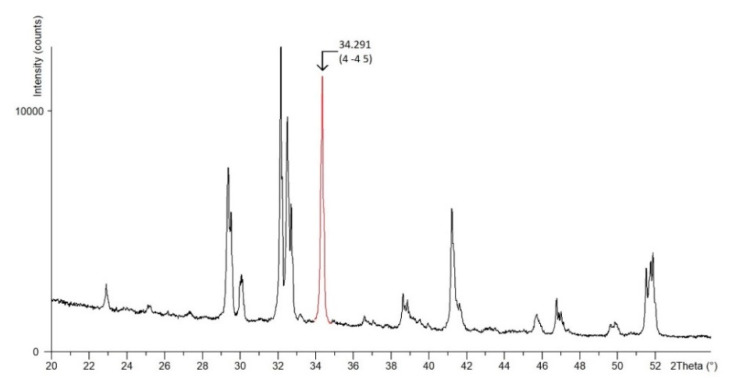
Detail of selected crystallographic plane $4\bar{4}5$ on X-ray pattern.

**Figure 5 materials-13-03734-f005:**
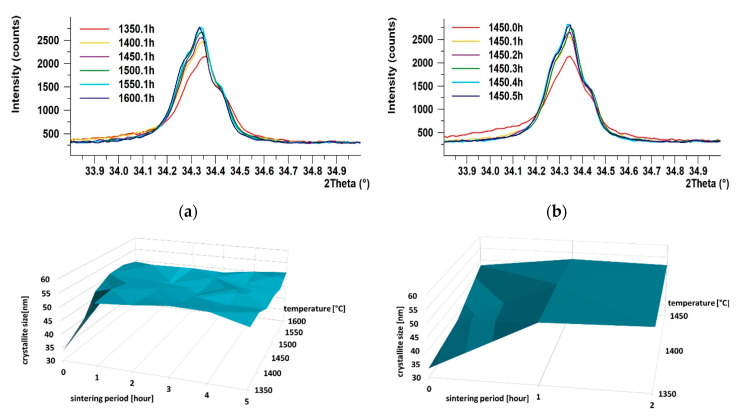
Development of crystallinity. (**a**) Influence of temperature on crystallite size at 1 h of sintering; (**b**) influence of sintering period on crystallite size at 1450 °C; (**c**) 3D graph of growth of crystallites in plane $4\bar{4}5$; (**d**) detail of 3D graph of growth of crystallites in plane $4\bar{4}5$.

**Figure 6 materials-13-03734-f006:**
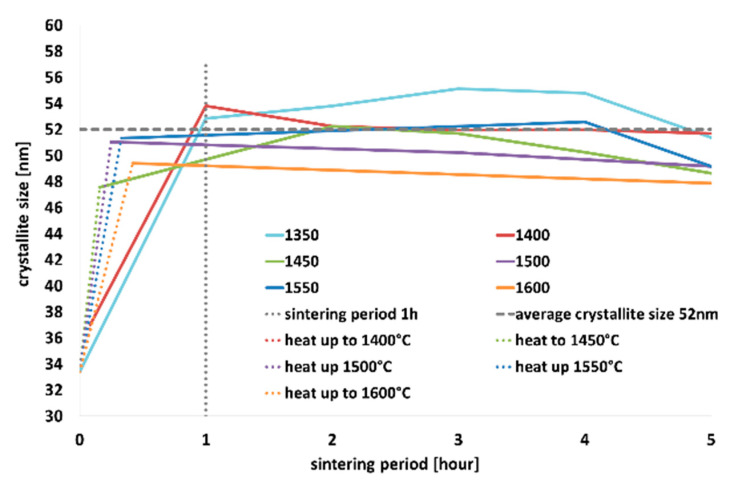
Influence of sintering period on crystallite size at investigated temperatures.

**Table 1 materials-13-03734-t001:** Chemical composition, in g, for 100 g of final T1 C_3_S product, C:S = 3.0 [27].

Component	CaCO_3_	SiO_2_
**Dosage [g]**	131.506	26.316

**Table 2 materials-13-03734-t002:** Summary of mineral composition detected using Rietveld method.

Temperature (°C)	Sintering Period (h)	T1–C_3_S (%)	β-C_2_S (%)	f-CaO (%)
1350	0	9.12	69.45	21.43
1350	1	68.08	24.28	7.66
1350	2	80.97	14.35	4.68
1350	3	84.36	11.91	3.73
1350	4	86.71	9.68	3.61
1350	5	89.37	7.76	2.87
1400	0	44.45	41.13	14.42
1400	1	83.27	12.36	4.37
1400	2	96.06	0.10	3.80
1400	3	95.81	2.01	2.18
1400	4	98.77	0.30	0.93
1400	5	97.06	1.50	1.44
1450	0	64.75	26.58	8.67
1450	1	91.60	6.14	2.26
1450	2	99.14	0.00	0.86
1450	3	98.80	0.50	0.70
1450	4	98.85	0.60	0.55
1450	5	98.65	0.40	0.95
1500	0	75.80	16.37	7.84
1500	1	98.89	0.00	1.11
1500	2	98.32	0.90	0.78
1500	3	99.35	0.00	0.65
1500	4	99.55	0.00	0.45
1500	5	99.45	0.00	0.55
1550	0	82.65	12.78	4.58
1550	1	98.34	0.80	0.86
1550	2	99.32	0.00	0.68
1550	3	99.00	0.40	0.60
1550	4	99.14	0.00	0.86
1550	5	99.22	0.00	0.78
1600	0	89.22	8.57	2.21
1600	1	99.22	0.00	0.78
1600	2	99.02	0.80	0.18
1600	3	98.92	0.30	0.78
1600	4	99.07	0.40	0.53
1600	5	99.37	0.00	0.63

**Table 3 materials-13-03734-t003:** Crystallite size in the plane $4\bar{4}5$.

Temperature [°C]	Crystallite Size [nm]					
	0 h	1 h	2 h	3 h	4 h	5 h
1350	33.391	52.239	53.813	55.128	54.793	51.371
1450	37.018	53.813	52.159	51.960	51.99	51.693
1450	47.562	52.259	51.693	50.263	49.652	49.652
1500	51.054	52.835	49.714	49.177	50.542	49.177
1550	51.342	52.593	49.177	49.987	48.652	47.886
1600	49.417	47.886	48.913	48.394	50.144	51.156

**Table 4 materials-13-03734-t004:** Growth rate of crystallite size.

Temperature (°C)	Growth Rate (nm/min)				
	0–1 h	1–2 h	2–3 h	3–4 h	4–5 h
1350	0.88	0.90	0.92	0.91	0.64
1400	0.97	0.95	0.94	0.95	0.69
1450	1.05	1.03	1.01	0.97	0.70
1500	1.11	1.10	1.09	1.12	0.76
1550	1.31	1.23	1.25	1.22	0.80
1600	1.37	1.40	1.38	1.41	0.90

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
