# Peer review of "Development of Crystallinity of Triclinic Polymorph of Tricalcium Silicate"

_materials, 2020, doi:10.3390/ma13173734_

Round 1
Reviewer 1 Report
The paper deals with crystallization process of triclinic tricalcium silicate during the firing process, which is of a great importance for clinker industry. Influence of temperature and sintering time on phase content and crystallite size were monitored in this respect by employing X-ray diffraction analysis and calculation of crystallinity using Scherrer’s equation. Overall, the paper brings interesting findings for the clinker industry and material science, and is recommendable for publishing.
There a few issues to fix in order to improve the manuscript final version:
- Page 1, line 31. Typo: The has been… (There has been…)
- Page 1, line 38: Grammar: …, thanks to its…(thanks to their).
- Page 2, line 59: Grammar: Jeffery has dealt… (Jeffery dealt…)
- Page 2, line 71: Grammar: … and its study…(and their study...).
- Page 2, line 89: Typo: …was 1000g… (was 1000 g….)
- Page 3, lines 94-98: Please, indicate what kind of furnace was used for sintering experiments.
- Page 3, line 100: λ=1.540598 (add Å unit)
- Page 3, line 106: Please, use diffraction line instead diffractive line.
- Page 3, line 107: ..The high diffraction line…(…A high-intensity diffraction line…)
- Page 3, line 109: It was determined (It is defined by…). Please, use proper Miller index assignment: minus sign should be over the number (Mind this throughout the text).
- Page 3, line 110: It should be “full width at half maximum” (make respective corrections throughout the text).
- Page 5, Figure 2 caption: Rephrase to: Graphical representation of the mineral composition change with increase of sintering period…
- Page 6, Figure 4: Figure 4e is missing! Rephrase the 4e caption to: Influence of temperature and sintering period on crystallite size.
- Page 7, Figure 5 caption: Rephrase the caption to: Influence of sintering period on crystallite size at various temperatures.
- Page 7, line 171: …lengthens. (…and its intensity increases.).
- Page 7, line 179, Typo: Omit comma at the end of the line.
- Page 7, lines 191-192: Rephrase to: ….the amount of alite increased relative to other minerals.
- Page 7, lines 195-196: Move the description of the amorphous phase determination method to the Chapter 2 (Materials and methods), lines 104-105.
- Page 8, line 205: Rephrase to: …, and significant peak overlapping of individual crystallographic planes…
Author Response
Thank you for your comments and fruitful discussion.
For all answers see the attached file please.

Reviewer 2 Report
This study reports the characterization of laboratory-prepared clinkers using X-ray diffraction with the aim of identifying the best firing temperature and duration in terms of alite yield and crystallite size. Therefore, I think this manuscript falls within the scope of Materials. I find it interesting and I appreciate the approach based on the characterization of standard materials under controlled conditions. I do not have comments about the methods and results. However, the Abstract, Introduction and Conclusions must be improved to emphasize the importance of this research, its place in relation to previous studies, and how it pushes forward the field of cement and concrete research. A lot of background information is given for granted and that belittles the value of this contribution. The research question is not clearly stated, nor are the broader implications of the results. I think the manuscript may be accepted for publication after the necessary revisions. Below I list detailed comments.
Line 8. An introductory sentence to the subject is necessary. Some people may be familiar with calcium silicate, but not with clinkers.
Line 10. This sentence, appropriately rearranged, could serve as introduction to the research question.
Line 18. The temperature part is unclear. Which temperature was used?
Lines 24-27. Again, the subject is not properly introduced. What are clinkers? Why is there a need to change fuel in their production? Why is their characterization important? It seems like the authors give for granted that readers are 100% familiar with clinkers, cement, and all of the problems related to their production.
Line 34. I would add a table listing all of the clinker minerals identified to date, where the most important ones are highlighted.
Line 40. Perhaps these methods should be briefly described.
Line 72. The research question behind this study and its aim is not clearly stated. See my comment for lines 24-27. Why is it important to characterize clinker composition? Why crystallite size matters? How does that affect cement and concrete research/production technology? What are the main contributions of this study (in one line)? Are there implications for other fields of research? I think the authors should develop more these points, otherwise the manuscript looks like a laboratory report.
Line 130. Is portlandite a result of cooling the samples in air?
Lines 185-185. This sentence can be improved. What are exactly the nodules mentioned by the authors and how do they affect the preparation process?
Line 192. This answers my previous question. The next step would be to try XRD with a heating stage.
Line 229. Why is the quality insufficient? This should be mentioned.
Line 239. I would add a short paragraph with the broader implications of these results for the field of cement research.
Author Response

(The authors gave the same response as above.)

Reviewer 3 Report
Although I am not a native speaker, I would suggest a few minor language corrections
line 9 "minerals" - I would suggest singular "mineral"
lines 15-16 either The effects ... were, or The effect ... was, rather not The effect ...were as it is in the manuscript
line 31 "The has been" looks unnatural, it ios not correct, I am afraid
line 61 Regourd, Guiniere and his coworkers. I would rather use "their coworkers"
lines 158-159 and Figure 4 (lines 162-165) for graphical representation see Fig. 4 a to e. Olthough Fig. 4e is explained in the lines 164-165 I cannot see this figure in the manuscript.
Author Response

(The authors gave the same response as above.)

Reviewer 4 Report
Dear Authors,
The manuscript ‘Development of the crystallinity of the triclinic polymorph of tricalcium silicate’ presents a range of results relevant to cement/concrete industry. The manuscript has a brief and good literature review and report interesting results, however there are some important aspects raised in the paper that’s should be clarified before the consideration for publication. Also the submission must be grammatically reviewed. See suggestions/comments below.
- English editing: For instance:
- Abstract, line 13: The article deals with the observing the crystallization process of triclinic tricalcium silicate during the firing process;
- Line 34 - About twenty clinker minerals are possible to identify in the clinker
- Lines 170 to 180 – full review
- Line 195: The amorphous phase content of the sample was tried to determine using the constant
- Discussion: full review
Therefore the entire submission needs a careful English editing/review.
- The calcination of raw materials for cement production usually uses temperatures around 900 °C. Also the sequence of alite transformations is shown in the diagram [4] of the manuscript shows the temperatures up to 1070C. How can the authors justify the temperature range used in the project and also the conclusion that triclinic alite obtained at 1350 °C is of insufficient quality. What are the implications of the results presented in this paper and the current practices in industry?
- Line 43 : Pt crucibles – add full description
- In the literature review, the authors cite several authors without citing the references, such as: Line 58 - Jeffery, Line 62 – Regourd and others,
- It would be interesting to add the XRD patterns for all tested mixes so the reader/reviewers can better understand the full XRD pattern of the alites procuded in this research.
- Line 139: Authors suggest that increasing firing temperature, the amount of triclinic alite increases linearly from 9.1 % at 1350 °C to 99.4 % at 1600 °C. Could the authors plot this trend and add to the paper?
- Line 193: Authos state that ‘Based on the calculation of the loss on ignition, portlandite was converted to free lime and the contents of the remaining minerals were increased by this value’. How can portlandite be converted to free lime? Was the alite hydrated in order to produce portlandite? How portlandite was converted back to free lime?
- Line 195: Authors state that amorphous phase content of the sample was tried to determine using the constant background intensity method where the crystallinity of a sample is defined as the intensity ratio of the diffraction peaks and of the sum of all measured intensity. The sentence need editing to clarify what the message to the readers. Also authors need to provide further explanation on how the amorphous content was quantified.
- Also authors state in line 198 that ‘determining the amorphous content was very complicated. The results showed that all raw materials reacted to the crystalline phase when it was reached 1350 °C.’ Again the sentence need English editing and also further explanation as the authors suggest in the conclusion that triclinic alite obtained at 1350 °C is of insufficient quality while the sentence states that 1350C is enough to promote all the crystalline reaction.
- Also in line 199 authors state that ‘we can state that the contents of individual minerals in the samples are not affected by the amorphous content.’
- Based on the comments I would suggest a full review of the discussions and conclusions. Also in the discussions, authors must specify which graph, table or figure their discussion is based on.
Author Response

(The authors gave the same response as above.)

Round 2
Reviewer 4 Report
Thanks for sending the new version of the manuscript.